# Industrial Anomaly Detection with Skip Autoencoder and Deep Feature Extractor

**DOI:** 10.3390/s22239327

**Published:** 2022-11-30

**Authors:** Ta-Wei Tang, Hakiem Hsu, Wei-Ren Huang, Kuan-Ming Li

**Affiliations:** 1Department of Mechanical Engineering, National Taiwan University, Taipei 10617, Taiwan; 23DFAMILY Technology Co., Ltd., New Taipei 23674, Taiwan

**Keywords:** anomaly detection, industrial defect detection, deep learning, feature extraction

## Abstract

Over recent years, with the advances in image recognition technology for deep learning, researchers have devoted continued efforts toward importing anomaly detection technology into the production line of automatic optical detection. Although unsupervised learning helps overcome the high costs associated with labeling, the accuracy of anomaly detection still needs to be improved. Accordingly, this paper proposes a novel deep learning model for anomaly detection to overcome this bottleneck. Leveraging a powerful pre-trained feature extractor and the skip connection, the proposed method achieves better feature extraction and image reconstructing capabilities. Results reveal that the areas under the curve (AUC) for the proposed method are higher than those of previous anomaly detection models for 16 out of 17 categories. This indicates that the proposed method can realize the most appropriate adjustments to the needs of production lines in order to maximize economic benefits.

## 1. Introduction

Owing to the developments in convolutional neural networks (CNN) over recent years, classical CNN models such as the ResNeXt [1], Dual Path Networks [2], and EfficientNet [3] have been proposed, which has been proven to possess good image recognition abilities. Consequently, it is great for researchers to continue working toward importing CNN models into inspection applications and obtain good results, such as for industrial inspection [4,5], cryo-electron tomogram classification [6], lithium-ion battery electrode defect detection [7], solar cell surface defect inspection [8], bolt joints monitoring [9] and rolling bearing robust fault diagnosis [10].

However, the training of CNNs requires many defect samples and necessitates additional manpower for defect labeling, which makes it difficult to import deep learning technology into actual production lines. Although many data augmentation methods have been proposed thus far [11,12,13,14], the application of CNNs in industrial testing remains limited.

As a solution to overcome this limitation, anomaly detection technology, an unsupervised learning method, has garnered significant research attention. In recent studies, anomaly detection was mainly realized using an autoencoder (AE) [15] or a Generative Adversarial Network (GAN) [16]. The principle involves training neural networks through good products to realize better feature reconstruction abilities for good products. With regard to production lines, an abnormal score is defined as the difference between reconstructed and original images or features. Anomaly detection methods such as AnoGAN [17], GANomaly [18], Skip-GANomaly [19], and DFR [20] have been proven to possess certain anomaly detection abilities. Their model architecture is shown in Figure 1.

Although these models have certain abilities in industrial anomaly detection, their poor feature extraction and reconstruction abilities limit their practical applications. Recently, scholars have attempted to overcome these problems by introducing skip connections [19], deep feature reconstruction algorithms [20], and multiple autoencoders [21]. In this paper, an anomaly detection model with a skip autoencoder and a deep feature extractor is proposed. This model was proven to possess better inspection ability than previous anomaly detection models for different datasets and under different conditions.

This paper mainly discusses the following issues regarding the proposed method:The MVTecAD, furniture wood, and mobile phone cover glass datasets for production lines were used to train and verify the proposed model, which was then compared with previous anomaly detection models.Different feature extractors were used to train the proposed model, and optimal feature extractor selection under different requirements was discussed.The proposed model was trained with different feature extract layers, and the corresponding effects were discussed.

## 2. Related Works

Over recent years, many excellent anomaly detection models have been proposed. In this study, four important anomaly detection models were selected for introduction. These models were trained and verified experimentally and then compared with the proposed method. The advantages and limitations of these models are shown in Table 1.

### 2.1. AnoGAN

AnoGAN [17] was the first deep learning algorithm that employed a GAN to detect anomalies. Its principle entails using a good product to train the model such that the network can generate good product images. During detection, the image that is most similar to the image to be measured is generated via an iterative method, and the defect score is defined based on the residual of the two images. However, this method requires a significant amount of computing resources because of the numerous iterations.

### 2.2. GANomaly

To address the problem of high computing resource consumption, Akcay et al. proposed an anomaly detection algorithm, GANomaly [18]. In addition to the concept of GAN-based anomaly detection, this approach employed an autoencoder as a generator. This autoencoder significantly reduces the computing resource consumption and time required for GANomaly. However, owing to the lack of the feature extraction and image reconstruction abilities, the model remains limited.

### 2.3. Skip-GANomaly

To overcome the problem of insufficient image reconstruction ability, Skip-GANomaly has been proposed [19]. The authors used the skip connection architecture to significantly enhance the image reconstruction ability. This model shows good detection ability for most categories in the MVTecAD dataset. However, in certain categories with more complex defects, the detection ability of Skip-GANomaly is limited, indicating that its ability to extract depth features warrants further improvements.

### 2.4. Deep Feature Reconstruction (DFR)

Deep feature reconstruction (DFR) [20] is an effective anomaly segmentation method that can inspect and segment anomalous images by applying a multi-scale regional feature generator. This generator can generate multiple representations from a pre-trained CNN model of an image, making it discriminative and beneficial for anomaly detection. By leveraging these descriptive regional features, the authors applied an efficient autoencoder and inspected the anomalous regions within images via fast feature reconstruction. This method is efficient and has considerable potential for practical applications. However, the feature reconstruction ability of the autoencoder in DFR is still unsuitable for detecting small defects. Therefore, the autoencoder of DFR should be improved.

## 3. Proposed Method

### 3.1. Model Architecture

The structure of the proposed method, as shown in Figure 2, comprises a pre-trained feature extractor and skip connection-based autoencoder. Inspired by DFR [20] and Skip-GANomaly [19], the feature extractor (FE) was designed as a pre-trained CNN (ResNeXt101 was used instead of VGG19 in the DFR) model. It can extract important features from an input image x. After feature extraction, these features are resized and concatenated as a three-dimensional tensor and imported to the autoencoder.

Furthermore, the proposed method uses the skip connection adopted in Skip-GANomaly. On importing this structure, the autoencoder (AE) in the proposed method is expected to realize better feature reconstruction than DFR. Furthermore, the introduction of this structure ensures that the model exhibits better stability and a higher anomaly detection ability.

During the training process, only good samples were input into the model. Therefore, the autoencoder in the model possessed better capability for reconstructing the characteristics of good products. During the detection process, provided the appropriate residual score is defined to express the characteristic tensor FE(x) and the tensor reconstructed via AE(FE(x)), the function of anomaly detection and segmentation can be realized.

Overall, the proposed method combines the advantages of DFR and Skip-GANomaly, which makes it suitable for feature extraction and reduction.

### 3.2. Training Process

In the training process, the training data x were inputed into the deep feature extractor, which was pre-trained using the ImageNet dataset [22]. Furthermore, the weighting of the feature extractor was locked during the training process. By being pre-trained with a large amount of image data, the feature extractor can effectively extract a significant part of the training data. This step allows the model to obtain more features in the training process and makes the model considerably better than the model trained using the image directly.

After feature extraction, the feature was resized and concatenated as a three-dimensional tensor and then imported to the skip-connect-based autoencoder. Furthermore, to ensure that the model has the best image reconstruction ability for good samples, the loss function, which is termed as contextual loss, was applied in model training. This loss function indicates the difference between the feature tensor FE(x) and tensor AE(FE(x)) reconstructed by the autoencoder.

Mathematically, distance L2 was used to clearly define the difference between these two tensors. Therefore, the contextual loss also employed this distance definition. It can be expressed as follows:(1)$$L=E_{x∼p_{x}}[||AE(FE\left(\left(x\right)\right)-{FE\left(x\right)||}_{2}]$$

The aim of training is to minimize contextual loss, which ensures that the model achieves the best feature reduction performance for normal samples and further improves the anomaly detection ability of the model.

### 3.3. Detection Process

In the detection phase, the image x to be tested is first inputted into the FE feature extractor (.) and is reshaped and concatenated as the feature tensor FE(x). Next, the feature tensor was input into the autoencoder for reconstruction, and the tensor AE(FE(x)) was output. The residual map, R(FE(x), AE(FE(x))), was used to calculate distance L2 for the two vectors. This can be expressed as:(2)$$R\left(FE\left(x\right),AE\left(FE\left(x\right)\right)\right)=||AE(FE\left(\left(x\right)\right)-FE\left(x\right)|{|}_{2}$$

As the proposed method only uses good products for training, it shows better feature reconstruction ability for good products. Therefore, the residual score in the residual map R(FE(x), AE(FE(x))) of the good products was lower than that of the bad products.

During detection, by adjusting different threshold values θ, the part where the residual score is greater than the threshold value is defined as the abnormal area.

## 4. Experimental Setup

### 4.1. Datasets

In this study, in addition to MVTec AD [23], which is widely used in anomaly detection, furniture wood and mobile phone glass-cover datasets on an actual production line were also applied to evaluate the detection ability of the proposed model. The following is an introduction to the three datasets.

#### 4.1.1. MVTec AD

MVTec AD [23] is currently one of the most widely used industrial datasets for anomaly detection [24,25,26,27,28,29]. The dataset contains 3629 training images and 1725 verification data points. It includes 15 categories: five of these correspond with texture detection items, whereas the remaining pertain to object detection items. These multiple types of components make MVTec AD more comprehensive in terms of evaluating the detection ability of a deep learning model. Sample data are shown in Figure 3.

#### 4.1.2. Production Line Smartphone Glass-Cover Dataset

This is a collection of smartphone cover glass data captured via a line-scan camera. It contains 200 pieces of mobile phone glass-cover. The images were scanned and divided into 329 training and 54 test images. In the testing data, there are 27 abnormal images, which contain various common types of defects on the production line. To balance the data, an equal number of normal images were added to the testing data.

Notably, smartphone glass-covers in a production line easily absorb dust, which can be conveniently eliminated. Therefore, the samples containing dust in this dataset are regarded as normal samples, which makes the detection of these data considerably more difficult.

#### 4.1.3. Production Line Furniture Wood Dataset

This is an anomaly detection dataset for furniture wood in an actual production line. It contains 3075 training and 740 test images, which were captured via a line scan camera. There are six tag categories in this dataset: normal products, holes, chalk, knots, and black. In the testing data, there are 370 abnormal images, including five types of this production line, and the number of each defect types is between 70 and 80. To balance the data, an equal number of normal images were added to the testing data. Compared to the wood dataset in MVTec AD, this dataset has more complex defect categories and characteristics, which are suitable for verifying the detection ability of deep learning models with difficult samples.

### 4.2. Training Process

During the development of the proposed method, in order to make the method better applied to real industrial inspection projects, we strive to find a set of hyperparameters that can have good detection ability in most cases. The learning rate is from 0.001 to 0.00001. In addition, the number of epochs was set from 50 to 500. Finally, it was observed that using Adam [30] as the optimizer, all test items can converge well in the case of 150 epochs and a learning rate of 0.0001. Therefore, in the experiment, this set of hyperparameters and optimizers was applied to minimize the loss function. In addition, different feature extraction backbones were applied, including MobileNet v3 small [31], MobileNet v3 large [31], VGG19 [20], resnext50 [1], and resnext101 [1]. Furthermore, the most suitable backbone under different conditions was discussed. In addition, there is a four-blocks hierarchy in ResNeXt 101. Higher-level blocks output semantic features, while lower-level blocks output image features. Therefore, the performance of different sets of output blocks is tested and discussed. Since the 4th block has a huge bias for the pre-trained dataset, only the first three blocks were used in the study. A system equipped with an Intel i7-10700k CPU and Nvidia RTX2080ti GPU was used to train and test the models in this study.

### 4.3. Evaluation Method

In this study, the performance of each model was evaluated using the area under the curve (AUC) of the receiver operating characteristics. The AUC can help in comprehensively evaluating the detection ability of a detector under various threshold values. It is also the most widely used evaluation method for anomaly detection and segmentation.

## 5. Experiment Results

### 5.1. MVTec AD Dataset

MVTec AD can be used to determine the detection ability of anomaly detection models in various projects. As listed in Table 2, the proposed method has the highest AUC for 14 of the 15 categories. Moreover, it is worth noting that the proposed method shows considerably better anomaly detection performance under four categories, including cables, metal nuts, tiles, and transformers, as compared with the previous detection methods. All these inspection categories have complex variations among the images, and the defects have irregular textures, which necessitates better feature extraction ability. Because AnoGAN, GANomaly, and Skip-GANomaly do not have a pre-trained feature extractor, they suffer from the problem of insufficient feature extraction ability when detecting these categories, thereby resulting in poor detection ability. However, although DFR has the architecture of a pre-trained feature extractor, the image reconstruction ability of its autoencoder is not qualified. Conversely, the advantages of the proposed method were remarkable because of its strong feature reconstruction ability and effective pre-trained feature extractor. The heat maps and segmented masks of the MVTec AD are shown in Figure 4. It can be observed from the figure that the proposed method can accurately segment the abnormal parts in the image.

### 5.2. Production Line Smartphone Glass-Cover and Furniture Wood Datasets

The smartphone glass-cover and furniture wood datasets can be used to test the anomaly detection ability of an anomaly detection model under actual detection environments. As summarized in Table 3, the proposed method features the best detection performance for both datasets. Dust is present on the good smartphone glass-cover products, whereas the furniture wood has more variable textures and defect types, both of which significantly increase the difficulty of data detection. Because the other four anomaly detection deep learning models do not have sufficient feature extraction and reconstruction capabilities, their disadvantages will be encountered when detecting such complex production line data. The heat maps and segmented masks of the proposed method are shown in Figure 4, indicating that the proposed method has significant anomaly detection and segmentation abilities.

### 5.3. Discussion of Inference Time

The inference times of AnoGAN, GANomaly, Skip-GANomaly, and the proposed method are shown in Table 4. It can be observed that the run time of AnoGAN is inappropriate for real-time anomaly detecting since the iterative algorithm. Although the inference times of GANomaly and Skip-GANomaly are faster than DFR and the proposed method, their AUC scores are inappropriate for industry cases. On the other hand, the proposed method and DFR can detect almost 100 pictures in one second and have excellent AUC scores. Therefore, they are most appropriate for production lines.

### 5.4. Discussion of Feature Extractor

The AUCs of different feature extractors, including MobileNet v3 Small, MobileNet v3 large, VGG19, ResNeXt50, and ResNeXt101, are listed in Table 5. Figure 5a shows the correlation between the average AUC and feature extractor. It is expected that the depth of the feature extractor is positively correlated with the AUC. However, it is notable that the average AUC of ResNeXt101 was only 0.07 higher than that of MobileNet v3 small. This feature enables the proposed method to be used effectively in edge-computing devices. Even if fewer computing resources are used, the proposed method can maintain good detection ability.

In addition, the use of feature extractor blocks is discussed. The correlation between the AUCs and block usage in ResNeXt101 is summarized in Table 6 and Figure 5b. It was observed that a greater number of blocks had a significant impact on the detection results. This is because the higher the number of blocks the neural network layer contains, the higher-dimensional the semantic information, which can be better used by the autoencoder. Therefore, if it is necessary to reduce computing resource consumption during actual detection cases, block 3 or a combination of blocks 2 and 3 can be selected to maintain a detection ability similar to that of the original model.

## 6. Conclusions

In this study, a new anomaly detection and segmentation model was proposed. This model is improved based on the DFR architecture. Its novelty and contributions are as follows. First, we use the skip-connection architecture to improve the feature reduction ability of the autoencoder in DFR. Second, we change the feature extractor from the VGG model to ResNeXt101 and explore the best combination of output blocks. Third, in addition to the opened dataset, two sets of production line data collected by our team are used to verify the performance of the model in actual industrial applications. With the skip connection and deep feature extractor, the proposed model exhibits good feature extraction and reconstruction abilities. Therefore, its performance for MVTec AD and two groups of production line anomaly detection datasets is significantly better than those of previous models. Furthermore, this study discussed the performance of the model in terms of computing resources. The results indicate that the proposed model can maintain good detection and segmentation abilities even if it is replaced with a lighter feature extractor. This implies that the proposed model shows good performance and is suitable for application in actual production line inspection tasks. However, the proposed method still has the following two limitations and challenges. First, since the model focuses on the detection of details, it is relatively poor in detecting macro-scale defects. Second, the method using deep learning will still take a long time to train. These are the directions for our team to study and improve in the future research.

## Figures and Tables

**Figure 1 sensors-22-09327-f001:**
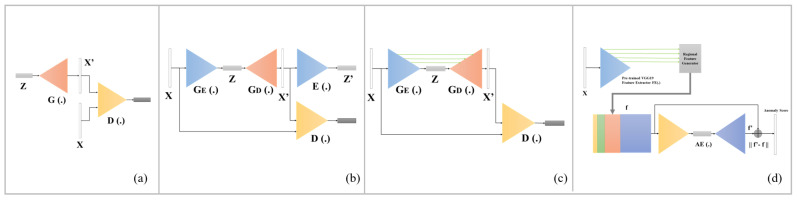
Architectures of: (**a**) AnoGAN, (**b**) GANomaly, (**c**) Skip-GANomaly, and (**d**) DFR.

**Figure 2 sensors-22-09327-f002:**
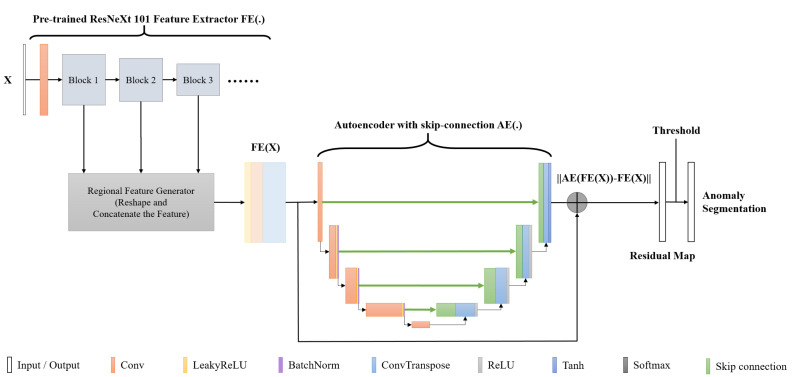
Structure of proposed method.

**Figure 3 sensors-22-09327-f003:**
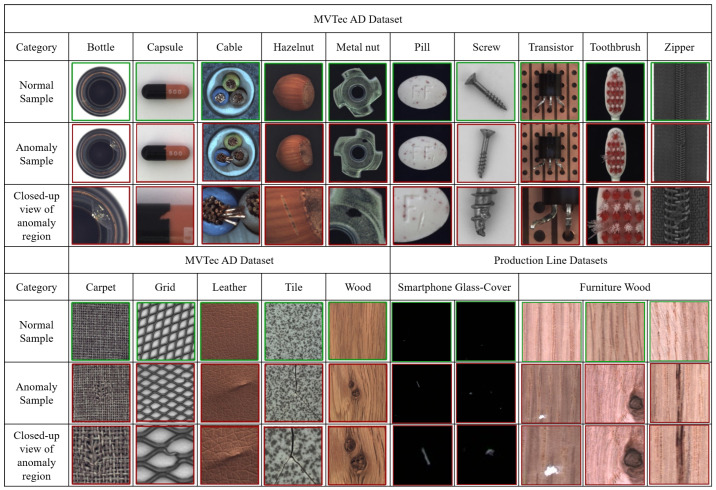
MVTec AD, production line smartphone glass-cover, and furniture wood datasets for industrial inspection.

**Figure 4 sensors-22-09327-f004:**
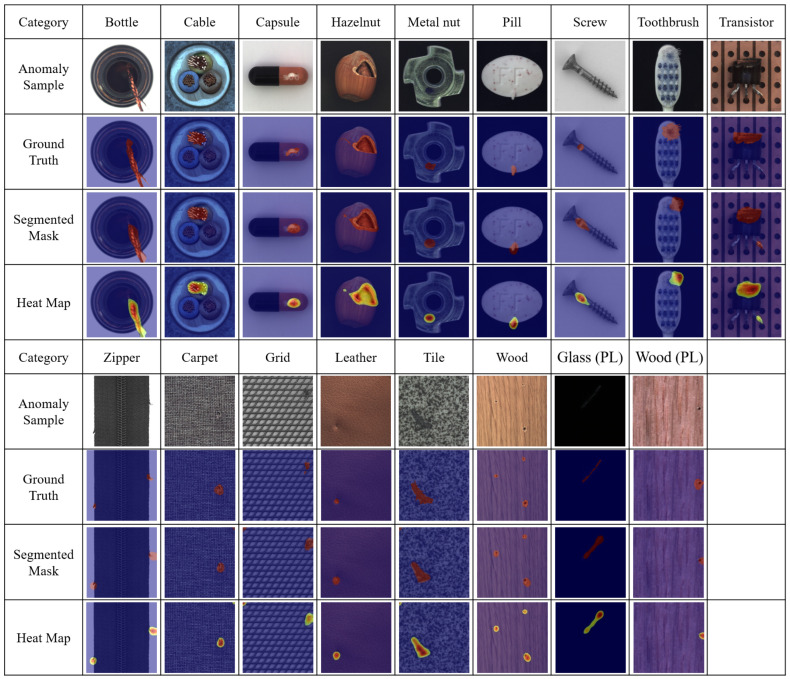
Segmented masks and heat maps of each category created using the proposed method.

**Figure 5 sensors-22-09327-f005:**
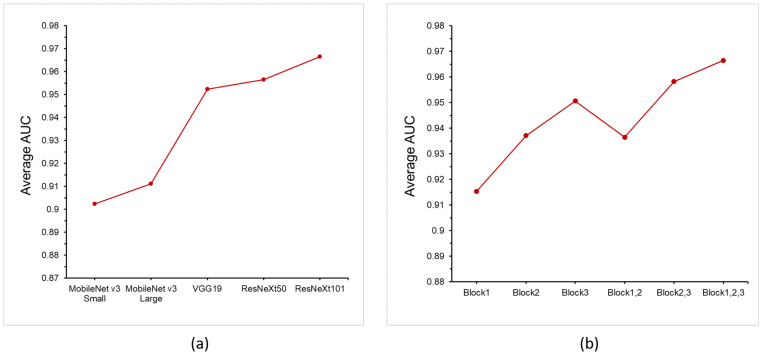
Average AUCs of all categories with: (**a**) different pre-trained feature extractor; (**b**) different blocks of ResNeXt101 feature extractor.

**Table 1 sensors-22-09327-t001:** Comparison of AnoGAN, GANomaly, Skip-GANomaly, and DFR.

	AnoGAN	GANomaly	Skip-GANomaly	DFR
Advantages	Training the model without abnormal data.	Much less inspection time than AnoGAN.	Better ability of feature reconstructing.	Great ability of feature extraction.
Limitations	Requires a significant amount of computing resources.	Feature extracting and reconstructing abilities are limited.	Cannot extract complex image feature.	Limited ability of feature reconstruction.

**Table 2 sensors-22-09327-t002:** AUCs of MVTec AD dataset using AnoGAN, GANomaly, Skip-GANomaly, and the proposed method.

Category	AnoGAN	GANomaly	Skip-GANomaly	DFR	Proposed Method
Bottle	0.82	0.82	0.91	0.94	0.98
Cable	0.77	0.64	0.65	0.87	0.95
Capsule	0.85	0.75	0.72	0.97	0.99
Carpet	0.55	0.83	0.52	0.98	0.98
Grid	0.59	0.89	0.85	0.96	0.91
Hazelnut	0.4	0.94	0.83	0.98	0.98
Leather	0.64	0.81	0.82	0.99	0.99
Metal nut	0.44	0.65	0.67	0.92	0.98
Pill	0.76	0.67	0.8	0.95	0.98
Screw	0.79	0.9	0.92	0.97	0.98
Tile	0.52	0.65	0.68	0.89	0.97
Toothbrush	0.88	0.85	0.78	0.97	0.99
Transistor	0.78	0.7	0.81	0.78	0.87
Wood	0.65	0.95	0.92	0.94	0.97
Zipper	0.77	0.67	0.67	0.95	0.98
Average	0.68	0.78	0.77	0.94	0.97

**Table 3 sensors-22-09327-t003:** AUCs of production line datasets using AnoGAN, GANomaly, Skip-GANomaly, and the proposed method.

Category	AnoGAN	GANomaly	Skip-GANomaly	DFR	Proposed Method
Glass (PL)	0.65	0.64	0.82	0.99	0.99
Wood (PL)	0.71	0.84	0.8	0.92	0.94
Average	0.68	0.74	0.81	0.96	0.97

**Table 4 sensors-22-09327-t004:** Inference Time of AnoGAN, GANomaly, Skip-GANomaly, and the proposed method.

	AnoGAN	GANomaly	Skip-GANomaly	DFR	Proposed Method
Time (ms)	7025	2.68	2.82	10.10	11.20

**Table 5 sensors-22-09327-t005:** AUCs of proposed method with different pre-trained CNN feature extractors.

Category	MobileNet (S)	MobileNet (L)	VGG19	ResNeXt50	ResNeXt101
Bottle	0.94	0.92	0.96	0.98	0.98
Cable	0.87	0.88	0.92	0.94	0.95
Capsule	0.94	0.94	0.98	0.99	0.99
Carpet	0.88	0.9	0.98	0.95	0.98
Grid	0.78	0.86	0.98	0.9	0.91
Hazelnut	0.95	0.95	0.98	0.98	0.98
Leather	0.98	0.98	0.98	0.98	0.99
Metal nut	0.87	0.95	0.94	0.98	0.98
Pill	0.92	0.92	0.97	0.96	0.98
Screw	0.92	0.95	0.98	0.98	0.98
Tile	0.98	0.95	0.91	0.97	0.97
Toothbrush	0.95	0.94	0.98	0.98	0.99
Transistor	0.74	0.66	0.79	0.82	0.87
Wood	0.93	0.94	0.95	0.97	0.97
Zipper	0.78	0.88	0.97	0.96	0.98
Glass (PL)	0.99	0.99	0.99	0.99	0.99
Wood (PL)	0.92	0.88	0.93	0.93	0.94
Average	0.9	0.91	0.95	0.96	0.97

MobileNet (S)–MobileNet v3 Small, MobileNet (L)–MobileNet v3 Large.

**Table 6 sensors-22-09327-t006:** AUCs of proposed method with different feature extractor blocks.

Category	Block1	Block2	Block3	Block1,2	Block2,3	Block1,2,3
Bottle	0.84	0.96	0.97	0.94	0.97	0.98
Cable	0.88	0.92	0.95	0.91	0.95	0.95
Capsule	0.93	0.97	0.98	0.96	0.98	0.99
Carpet	0.93	0.92	0.95	0.94	0.96	0.98
Grid	0.88	0.91	0.87	0.92	0.9	0.91
Hazelnut	0.97	0.96	0.97	0.97	0.98	0.98
Leather	0.98	0.99	0.97	0.98	0.97	0.99
Metal nut	0.96	0.96	0.95	0.96	0.97	0.98
Pill	0.92	0.95	0.97	0.95	0.98	0.98
Screw	0.97	0.97	0.96	0.97	0.97	0.98
Tile	0.97	0.97	0.95	0.97	0.96	0.97
Toothbrush	0.92	0.96	0.98	0.96	0.98	0.99
Transistor	0.66	0.71	0.88	0.71	0.87	0.87
Wood	0.96	0.96	0.94	0.96	0.96	0.97
Zipper	0.92	0.94	0.96	0.94	0.97	0.98
Glass (PL)	0.99	0.99	0.99	0.99	0.99	0.99
Wood (PL)	0.88	0.89	0.92	0.89	0.93	0.94
Average	0.92	0.94	0.95	0.94	0.96	0.97

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
