# Peer review of "Industrial Anomaly Detection with Skip Autoencoder and Deep Feature Extractor"

_sensors, 2022, doi:10.3390/s22239327_

Round 1

Reviewer 1 Report

This paper proposes a new anomaly detection and segmentation model which combines the advantages of the DFR and Skip-GANomaly. With the skip connection and deep feature extractor, the proposed model exhibits good feature extraction and reconstruction abilities. My comments are as follows:

1. The investigating subject is widely studies in the literature, but the literature review section is insufficient. The authors should present a comprehensive literature review on technological developments. Some latest references can be referred to the paper for improving the review part. For example, Doi: 10.1177/14759217221088492; Doi: 10.1177/14759217221091131.

2. How to select appropriate hyperparameters for the deep learning model, such as learning rate, optimizer type, etc?

3. In this paper, the authors discuss the use of different blocks in ResNeXt101 and their effects on the detection results. However, block1, block2 and block3 are not explained in this paper. Please give a brief description of different blocks.

4. The authors should highlight the contributions of the work in conclusion section and explain the novelty of this proposed model in comparison with the previous studies.

Author Response

Response to Reviewer 1’s Comments

Manuscript ID: sensors-2047302

Title: Industrial Anomaly Detection with Skip Autoencoder and Deep Feature Extractor

Author: Ta-Wei Tang , Hakiem Hsu , Wei-Ren Huang , Kuan-Ming Li *

Thank you very much for the Editor’s letter. Also, the authors would like to thank the referees for the valuable comments. The following changes or corrections are made in the revised manuscript according to the referees' comments:

Reviewer’s comments:

Point 1. The investigating subject is widely studies in the literature, but the literature review section is insufficient. The authors should present a comprehensive literature review on technological developments. Some latest references can be referred to the paper for improving the review part. For example, Doi: 10.1177/14759217221088492; Doi: 10.1177/14759217221091131.

Response 1.  Thanks for reviewer’s comment. We have referred these two papers in the Introduction Section.

Point 2.  How to select appropriate hyperparameters for the deep learning model, such as learning rate, optimizer type, etc?

Response 2.  Thanks for reviewer’s comment. We have added the following sentence “During the development of the proposed method, in order to make the method better applied to real industrial inspection projects, we strive to find a set of hyperparameter that can have good detection ability in most cases. The learning rate from 0.001 to 0.00001. In addition, the number of epochs was set from 50 to 500. Finally, it was observed that using Adam [30] as the optimizer, all test items can converge well in the case of 150 epochs and a learning rate of 0.0001. Therefore, in the experiment, this set of hyperparameter and optimizer was applied to minimize the loss function.” to section 4.2 to describe this issue.

Point 3.  In this paper, the authors discuss the use of different blocks in ResNeXt101 and their effects on the detection results. However, block1, block2 and block3 are not explained in this paper. Please give a brief description of different blocks.

Response 3.  Thanks for reviewer’s comment. We have added “Also, there are 4 blocks hierarchy in ResNeXt 101. Higher-level blocks output semantic features, while lower-level blocks output image features. Therefore, the performance of different set of output blocks are test and discussed. Since the 4th block have huge bias for the pre-trained dataset, only the first three blocks were used in the study.” to section 4.2 to describe this issue.

Point 4.  The authors should highlight the contributions of the work in conclusion section and explain the novelty of this proposed model in comparison with the previous studies.

Response 4.  Thanks for reviewer’s comment. We have added “This model is improved based on the DFR architecture. Its novelty and contributions are as follows. First, use the skip-connection architecture to improve the feature reduction ability of the auto-encoder in DFR. Second, change the feature extractor from VGG model to ResNeXt101 and explore the best combination of output blocks. Third, in addition to the opened dataset, two sets of production line data collected by our team are used to verify the performance of the model in actual industrial applications.” to Conclusions to describe this issue.

Reviewer 2 Report

 In this paper, the authors presented a novel deep learning model for anomaly detection to overcome this bottleneck. Leveraging a powerful pre-trained feature extractor and the skip connection, the proposed method achieves better feature extraction and image restoration capabilities. Results reveal that the areas under the curve (AUC) for the proposed method are higher than those of previous anomaly detection models for 16 out of 17 categories, as the author claimed.

The paper needs extensive revisions before considering for publication, such as:

-        The novelty is completely not clear. As noticed, the applied method all existing methods, I did not see any new method developed. Or, maybe the description of the contribution is not clear. So, clarify this issue.

-        The complexity and computation time should be studied and discussed.

-        What about the dividing of the training and testing samples? Did you test the model with completely unseen samples?

-        More details about the model are need. For example, pseudocodes, or source codes.

-        Limitations and challenges must be discussed.

-        A proof reading is also needed.

Author Response

Response to Reviewer 2’s Comments

Manuscript ID: sensors-2047302

Title: Industrial Anomaly Detection with Skip Autoencoder and Deep Feature Extractor

Author: Ta-Wei Tang , Hakiem Hsu , Wei-Ren Huang , Kuan-Ming Li *

Thank you very much for the Editor’s letter. Also, the authors would like to thank the referees for the valuable comments. The following changes or corrections are made in the revised manuscript according to the referees' comments:

Reviewer’s comments:

Point 1. The novelty is completely not clear. As noticed, the applied method all existing methods, I did not see any new method developed. Or, maybe the description of the contribution is not clear. So, clarify this issue.

Response 1.  Thanks for reviewer’s comment. We have added “This model is improved based on the DFR architecture. Its novelty and contributions are as follows. First, use the skip-connection architecture to improve the feature reduction ability of the auto-encoder in DFR. Second, change the feature extractor from VGG model to ResNeXt101 and explore the best combination of output blocks. Third, in addition to the opened dataset, two sets of production line data collected by our team are used to verify the performance of the model in actual industrial applications.” to Conclusions to describe this issue.

Point 2.  The complexity and computation time should be studied and discussed.

Response 2.  Thanks for reviewer’s comment. We have added the sub section “5.3. Discussion of Inference Time” to describe this issue.

Point 3.  What about the dividing of the training and testing samples? Did you test the model with completely unseen samples?

Response 3.  Thanks for reviewer’s comment. All of the models are tested by completely unseen samples. MVTecAD is opened datasets, the training and testing samples are divided by the author of the datasets. Also, we have added “In the testing data, there are 27 abnormal images, which contain various common types of defects on the production line. To balance the data, an equal number of normal images were added to the testing data.” to section 4.1.2 and added “In the testing data, there are 370 abnormal images, including five types of this production line, and the number of each defect types is between 70-80. To balance the data, an equal number of normal images were added to the testing data.” to section 4.1.3 to describe the dividing method of other two datasets.

Point 4.  More details about the model are need. For example, pseudocodes, or source codes.

Response 4.  Thanks for reviewer’s comment. However, according to the industry-university cooperation agreement, this research code will belong to our cooperating company. Therefore, we cannot release the code until the related product has been released for more than 2 years. However, we will negotiate with the company to release the code as soon as possible. Thanks for your understanding.

Point 5.  Limitations and challenges must be discussed.

Response 5.  Thanks for reviewer’s comment. We have added “However, the proposed method still has the following two limitations and challenges. First, since the model focuses on the detection of details, it is relatively poor in detecting macro-scale defects. Second, the method using deep learning will still take a long time to train. These are the directions for our team to study and improve in the future research.” to Conclusions to describe this issue.

Point 6.  A proof reading is also needed.

Response 6.  Thanks for reviewer’s comment. We have carefully proofread the entire article and corrected grammatical and wording errors.

Round 2

Reviewer 2 Report

This version was improved it can be accepted for publication.